# Optimizing laboratory-based surveillance networks for monitoring multi-genotype or multi-serotype infections

Qu Cheng[1], Philip A. Collender[1], Alexandra K. Heaney[1], Aidan McLoughlin[2], Yang Yang[3], Yuzi Zhang[4], Jennifer R. Head[5], Rohini Dasan[1], Song Liang[6], Qiang Lv[7], Yaqiong Liu[8], Changhong Yang[9], Howard H. Chang[4], Lance A. Waller[4], Jon Zelner[10,11], Joseph A. Lewnard[5], Justin V. Remais[1]*

1 Division of Environmental Health Sciences, School of Public Health, University of California, Berkeley, Berkeley, California, United States of America, 2 Division of Biostatistics, School of Public Health, University of California, Berkeley, Berkeley, California, United States of America, 3 College of Public Health and Health Professions and Emerging Pathogens Institute, University of Florida, Gainesville, Florida, United States of America, 4 Department of Biostatistics and Bioinformatics, Rollins School of Public Health, Emory University, Atlanta, Georgia, United States of America, 5 Division of Epidemiology, School of Public Health, University of California, Berkeley, Berkeley, California, United States of America, 6 Department of Environmental and Global Health College of Public Health and Health Professions, University of Florida, Gainesville, Florida, United States of America, 7 Institute of Health Informatics, Sichuan Center for Disease Control and Prevention, Chengdu, Sichuan, People's Republic of China, 8 Institute of Acute Infectious Disease Control and Prevention, Sichuan Center for Disease Control and Prevention, Chengdu, Sichuan, People's Republic of China, 9 Division of Business Management and Quality Control, Sichuan Center for Disease Control and Prevention, Chengdu, Sichuan, People's Republic of China, 10 Department of Epidemiology, School of Public Health, University of Michigan, Ann Arbor, Michigan, United States of America, 11 Center for Social Epidemiology and Population Health, School of Public Health, University of Michigan, Ann Arbor, Michigan, United States of America

* jvr@berkeley.edu

**Data Availability Statement:** All relevant data are available from https://github.com/qu-cheng/Lab_surveillance_optimization.

## Abstract

With the aid of laboratory typing techniques, infectious disease surveillance networks have the opportunity to obtain powerful information on the emergence, circulation, and evolution of multiple genotypes, serotypes or other subtypes of pathogens, informing understanding of transmission dynamics and strategies for prevention and control. The volume of typing performed on clinical isolates is typically limited by its ability to inform clinical care, cost and logistical constraints, especially in comparison with the capacity to monitor clinical reports of disease occurrence, which remains the most widespread form of public health surveillance. Viewing clinical disease reports as arising from a latent mixture of pathogen subtypes, laboratory typing of a subset of clinical cases can provide inference on the proportion of clinical cases attributable to each subtype (i.e., the mixture components). Optimizing protocols for the selection of isolates for typing by weighting specific subpopulations, locations, time periods, or case characteristics (e.g., disease severity), may improve inference of the frequency and distribution of pathogen subtypes within and between populations. Here, we apply the Disease Surveillance Informatics Optimization and Simulation (DIOS) framework to simulate and optimize hand foot and mouth disease (HFMD) surveillance in a high-burden region of western China. We identify laboratory surveillance designs that significantly outperform the existing network: the optimal network reduced mean absolute error in estimated

**Funding:** This research was supported in part by the National Institutes of Health (R01AI125842), the National Science Foundation (DEB 2032210), and by the MIDAS Coordination Center (MIDASSUP2020-4) by a grant from the National Institute of General Medical Science (3U24GM132013-02S2) to JVR. The funders had no role in study design, data collection and analysis, decision to publish, or preparation of the manuscript.

**Competing interests:** The authors have declared that no competing interests exist.

serotype-specific incidence rates by 14.1%; similarly, the optimal network for monitoring *severe* cases reduced mean absolute error in serotype-specific incidence rates by 13.3%. In both cases, the optimal network designs achieved improved inference without increasing subtyping effort. We demonstrate how the DIOS framework can be used to optimize surveillance networks by augmenting clinical diagnostic data with limited laboratory typing resources, while adapting to specific, local surveillance objectives and constraints.

## Author summary

Laboratory-based tests can determine the specific agents that cause infectious diseases, providing important information for disease surveillance, and helping to understand the transmissibility, clinical spectrum, evolutionary trends, and subtype-specific risk factors of infections caused by pathogens with multiple types. However, pathogen typing is relatively expensive and scarce, and thus there is widespread interest in the optimal allocation of laboratory typing resources in the design of disease surveillance systems, even as such surveillance optimization methods have been understudied. Here we apply the Disease Surveillance Informatics Optimization and Simulation (DIOS) framework to the problem of optimal allocation of laboratory-typing within clinical surveillance systems. We develop methods for optimizing allocation of laboratory-typing across locations and clinical subgroups (e.g., severe vs. mild cases), and demonstrate the approach using real-world data from a surveillance network monitoring Hand Foot and Mouth Disease in western China. Using a series of simulation-optimization studies, we identified surveillance networks that are capable of reducing the mean absolute error of serotype-specific incidence rates by 13.3% among severe cases, and 14.1% among all cases. The methods demonstrated here are but one of many approaches through which the DIOS framework could be utilized to better leverage laboratory-typing infrastructure to track pathogen-specific epidemiologic trends.

## 1 Introduction

Laboratory procedures to identify pathogen subtypes (e.g., with respect to strain, genotype, serotype, variant, or phenotypic traits such as drug resistance) are important components of infectious disease surveillance, yielding information on transmissibility, clinical spectrum, evolutionary trends, and subtype-specific risk factors [1–7]. Indeed, information gathered from laboratory pathogen typing is integral to modern disease surveillance, enabling the discovery of SARS-CoV-2 variants with higher transmissibility [7], influenza A serotypes with high mortality and transmissibility [5], changes in the prevalence rate of drug-resistant tuberculosis and Methicillin-resistant Staphylococcus aureus (MRSA) [8,9], shifts in dominant serotypes causing invasive pneumococcal disease[6], and differing routes of infection across hepatitis C virus genotypes [10].

Such findings can guide the development, allocation, and evaluation of public health interventions. For instance, knowledge about the relative prevalence and virulence of pathogen subtypes is used to prioritize subtypes for vaccine or treatment development [11–13]; identify high-risk subpopulations to target with interventions [14]; and evaluate the risk of unintended consequences of interventions, such as serotype replacement [15,16]. Because of the high cost and complexity of collecting and processing laboratory samples, and because data on pathogen subtype may not inform clinical decision-making for individual patients, typing is often

undertaken for only a small subset of clinical cases. As examples, 2.8% of COVID-19 cases in the United States have been sequenced since January 10, 2020 [17]; <3% of hand foot and mouth disease (HFMD) cases in China were serotyped between 2011 and 2015 [2]; and only 9 influenza cases per participating laboratory are required to be characterized every other week across the United States to evaluate whether circulating influenza viruses are sufficiently similar genetically and/or antigenically to those that are included in current influenza vaccines [18].

Subtyping even a small proportion of cases may enable relevant inferences about the distribution of pathogen subtypes of interest within the larger set of clinically identified cases of a disease. However, in the absence of well-designed protocols for selection of isolates for subtyping, direct extrapolation of data from subtyped cases to the much broader population of clinical cases is susceptible to substantial biases, e.g., laboratory typing tends to be affected by clinical severity, healthcare capabilities, case clustering status, seasonality and other factors. In China, for example, severe cases of HFMD were serotyped at a rate of 72%, but only 2% of mild cases were serotyped [2].

Such imbalanced sampling regimes, often arising from practical clinical considerations, can substantially impact estimates of genotype-, serotype-, or other subtype-specific epidemiologic parameters (e.g., subtype-specific incidence; response of pathogen subtype distribution to public health interventions; etc.) [2]. Statistical inference may be improved by modifying sampling design to minimize such biases across the surveillance network, such as by redistributing total samples across time, space, or populations. In practice, sampling designs for laboratory subtyping vary widely across surveillance systems, and are generally *ad hoc* in nature, constrained by budget, logistics, or infrastructure [2,4]. Optimizing sampling under these constraints is a high priority for laboratory surveillance systems [2,19].

Here, we develop methods to support the optimization of sampling clinical cases for laboratory typing with the goal of improved monitoring of the distribution of specific pathogen subtypes, while abiding by constraints on available resources, e.g., the total number of clinical cases subjected to subtyping. Our work is based on the Disease Surveillance Informatics Optimization and Simulation (DIOS) framework [20], which iteratively evaluates surveillance network performance on predefined goals while varying surveillance system design parameters using numerical optimization algorithms. We adapt the DIOS framework to the problem of optimal allocation of laboratory typing resources across subregions and case severity groups of a surveillance network in order to minimize error in estimating the incidence rates of pathogen subtypes causing a clinically-diagnosed disease. We examine major enteroviruses causing HFMD in a region experiencing a high HFMD burden in China to illustrate the application of this framework.

## 2 Materials and methods

### 2.1 General framework for optimizing laboratory-based surveillance systems to monitor multi-genotype or multi-serotype infections

**Simulation framework.** DIOS [20] is a simulation-based optimization framework to facilitate the design of robust disease surveillance systems. DIOS functions by linking disease system models that simulate epidemiologic processes with surveillance system models that simulate information derived from alternative surveillance system designs. Applying DIOS involves specifying surveillance objectives (e.g., accurate estimation of disease frequency; timely outbreak detection; accurate estimation of intervention effectiveness), defining relevant surveillance design parameters (e.g., target population, diagnostic techniques, and site enrollment), and imposing operational constraints (e.g., total resources available for laboratory typing) (Box 1).

Box 1. Example DIOS optimization procedure

Consider the problem of identifying the optimal active surveillance strategy to estimate the incidence rate of a disease in key subpopulations, with possible designs given by altering the number of individuals to be surveyed across each subpopulation and the diagnostic test.

*Objective*: minimize bias in estimated incidence rate within each subpopulation

*Design parameters*: 1) number of persons to be selected from each subpopulation for diagnostic testing; and 2) laboratory technique used for diagnostic testing (e.g., polymerase chain reaction test, rapid antigen test, and culture)

*Models*: The disease system model simulates the underlying dynamics of the target disease in each subpopulation. The surveillance model selects a given number of persons from each subpopulation for testing according to the current design parameter values, simulates test results according to the sensitivity, specificity, or any other relevant characteristics of the test, and extrapolates incidence rates from the test results. The performance of the surveillance model is then evaluated by how close estimated incidence rates are, on average, to the true values simulated by the disease system model, using a score such as mean absolute error. After each evaluation, an optimization search algorithm (e.g., simulated annealing; evolutionary algorithm; particle swarm optimization) is used to update the design parameter, possibly based on an archive of previous performance including the current iteration. The following process is repeated:

1. propose a new design →

2. simulate disease and surveillance processes →

3. evaluate performance of design →

until a stopping criterion is met, such as exceeding a preset computational budget or failing to improve upon the best simulated design for a certain number of iterations.

The design parameters associated with the best performance are returned (see [20]).

The disease system model (see [20]) may be statistical, mechanistic, or an ensemble of different models or parameters that account for epistemic and parametric uncertainties, and should be developed with special attention to representing any processes thought to be relevant to the surveillance process. Multiple realizations of the disease system model, which may comprise incident cases or other phenomena of interest, is then filtered through measurement processes simulated by the surveillance model [20], which mimics relevant data collection and processing behaviors of a surveillance system, subsequently yielding estimates of the target epidemiologic parameter(s) (e.g., disease incidence; probability of an outbreak; change in incidence following intervention) that can be compared to true underlying values generated by the disease system model to assess the performance of the surveillance design.

## Adaptation of DIOS to the design of laboratory-based surveillance for monitoring infections caused by multiple genotypes or serotypes

To apply the DIOS framework to the optimization of surveillance for multiple pathogen subtypes (Fig 1), a first step is to define objective functions to evaluate surveillance performance

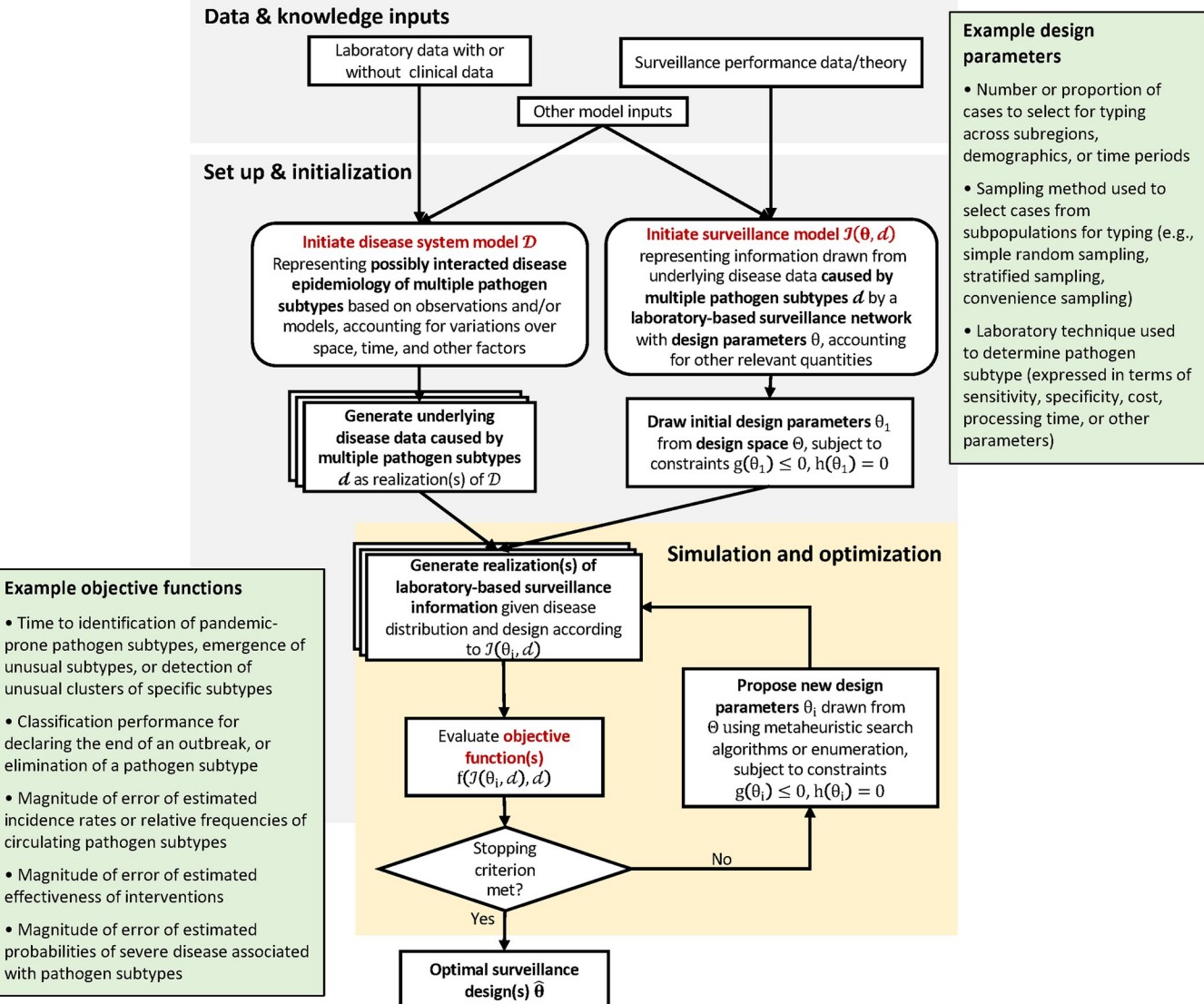

**Fig 1. Schematic of the DIOS framework for optimizing surveillance of infections caused by multiple pathogen subtypes, with example design parameters and objective functions presented in green boxes.**

on estimating epidemiologic parameter(s) related to pathogen subtype(s) of interest. For instance, researchers may be interested in early detection of a more infectious variant of a circulating infection, e.g., the Delta variant of SARS-CoV-2, and therefore specify an objective as minimizing prevalence of that subtype by the time it is detected. If the overall composition of cases associated with multiple pathogen subtypes is of interest, a suitable objective might be to minimize the mean absolute error of incidence rate estimates across subtypes. More example objective functions can be found in Fig 1.

Second, design parameters relevant to laboratory surveillance must be conceptualized and defined in the surveillance system model. Examples of surveillance design parameters that may bear on the abovementioned objectives include the number of cases sampled for typing across different subpopulations; the sampling protocols used to select cases to subtype from these subpopulations; and the laboratory techniques used to identify pathogen subtypes.

Third, the disease system model must represent the dynamics of multiple pathogen sub-types and their possible interactions, and be able to correct for known biases in the observed surveillance data. For example, the negative interaction between dengue virus serotypes—possibly due to short-term cross-protection [21]—would need to be accounted for in a disease model simulating the incidence of dengue fever associated with multiple serotypes. Similarly, any tendency to select severe cases for typing would need to be corrected by incorporating the heterogenous selection probability of different disease severity groups in the disease system model [2,4].

Finally, the DIOS surveillance model must be able to represent necessary characteristics of laboratory-based surveillance systems, such as assay-dependent classification performance, turnaround time, or cost. For instance, if the design parameter subject to optimization is the laboratory technique used to determine the presence of a pathogen subtype, the surveillance model should be able to simulate known relevant attributes of the candidate techniques, such as the probability of false positive or false negative results.

## 2.2 Application of DIOS to optimize laboratory-based surveillance of serotypes of enteroviruses causing HFMD

**2.2.1 Background.** HFMD is a pediatric infectious disease of growing public health importance [22,23], with a particularly high burden in East and Southeast Asia [22,24]. A variety of enteroviruses transmitted through fecal-oral or respiratory routes are causative agents of HFMD—including enterovirus-A71 (EV-A71), coxackievirus-A16 (CV-A16), CV-A6, and CV-A10 [25]. EV-A71 and CV-A16 have long been the serotypes associated with the highest disease burden, but other serotypes, such as CV-A6 are emerging with increasing clinical relevance in recent years [26,27]. The specific etiology of HFMD impacts the severity of symptoms, and has ramifications for intervention strategies, particularly vaccination. In China, recent deployment of monovalent vaccines against EV-A71, the most virulent serotype, has led to a reduction in the incidence of severe HFMD, but the overall incidence of HFMD is still rising, suggesting the possibility of serotype replacement [15]. Thus, it is critical to optimize laboratory surveillance to accurately estimate incidence of all HFMD and severe HFMD attributable to various enterovirus serotypes within the constraints of available resources.

**2.2.2 Study region and surveillance system.** Between 2004–2013, HFMD was the leading cause of death for children under five years old in China amongst all 39 nationally notifiable infectious diseases, and had the highest incidence of any infectious disease in the country [28, 29]. Since the inclusion in 2008 of HFMD on the list of mandatory notifiable infectious diseases in China, over 22.5 million cases have been reported across the country as of 2019 [30]. Sichuan Province (population >80 million) exhibits strong spatial and temporal heterogeneity in HFMD disease burden across prefectures, and is among multiple ongoing centers of transmission [31]. Clinically diagnosed HFMD cases are registered by the National Infectious Disease Reporting System (NIDRS), which covers nearly all healthcare facilities in China [32]. Clinical cases of HFMD are diagnosed by the presence of papular or vesicular rash on hands, feet, mouth or buttocks with or without fever, and are required to be reported to NIDRS within 24 hours [23]. Because of the narrow affected age group, distinct clinical features, and known seasonality of the disease, clinical diagnosis is considered to be highly specific [33].

Specimens are collected from a subset of clinical cases presenting at sentinel hospitals in an *ad hoc* manner to determine the underlying serotype using reverse-transcriptase polymerase chain reaction (RT-PCR), and the test results are reported to a laboratory surveillance system [31]. Deidentified data on clinical HFMD cases were obtained for the 21 prefectures of Sichuan from Sichuan Center for Disease Control and Prevention, including serotype information

(recorded as EV-A71, CV-A16, or other enterovirus) and indicators of case severity, and were aggregated at the prefectural level for each year from 2009 up to 2015, stopping one year before the introduction of EV-A71 vaccines into the region in 2016 [15]. Prefecture-level population data were collected from public sources for 2009–2015 [34].

The epidemiologic data supporting the optimization analysis herein included a total of 388,365 HFMD cases reported from 2009–2015 in Sichuan, of which 0.87 percent (3,380 cases) were severe. Annual HFMD incidence rates increased gradually over time (Fig 2A) and varied substantially across space (Fig 2B), with the highest annual mean incidence rate observed in Chengdu, the capital prefecture, and its surrounding prefectures, as well as the city with the highest per capita gross regional product, Panzhihua, in the southwest of the province. Laboratory tests were conducted for 22,100 cases (5.7%), with 52% of severe cases and 5.3% of mild cases subjected to serotyping. The number of laboratory-tested cases increased over time (Fig 2C) and exhibited substantial spatial variation (Fig 2D). The proportion of all, mild, and severe HFMD cases tested from 2009–2015 by prefecture are shown in S1 Fig. CV-A16, EV-A71, and other enteroviruses caused 26.6%, 29.1% and 44.3% of all serotyped cases, and 7.3%, 58.5% and 34.1% of severe serotyped cases, respectively, indicating EV-A71 (CV-A16) tended to cause severe (mild) symptoms. CV-A6 and CV-A10 likely constitute the majority of other enteroviruses in circulation [35–38].

**2.2.3 Defining the optimization problem.** We pursued optimization of estimates of total and severe HFMD incidence across serotypes, with the proportion of typing allocated to each prefecture ("location") and case severity group (mild and severe) as design parameters. The optimization seeks the sample allocation vector $\boldsymbol{\theta} = \{\theta_1, \theta_2,...,\theta_I, \theta_s\}$ ($I = 21$) that minimizes the mean absolute error (MAE) in the estimates of serotype-specific incidence rate of: 1) total; and 2) severe HFMD across time, space, and realizations, where $\theta_i$ represents the proportion of total serotyping resources allocated to the $i$-th location in the study province, and $\theta_s$ represents the probability of a severe case being tested, which is assumed to be fixed across locations. After allocating typing resources to severe cases as defined by $\theta_s$, the remaining available typing according to $\theta_i$ at location $i$ will be allocated to mild cases. The total number of cases sampled for subtyping each year is fixed at the observed frequency of typing (Fig 2C). The optimization problem can be formalized as:

$$\underset{\boldsymbol{\theta}}{\text{minimize }} f_n(\boldsymbol{\theta})$$

$$\text{subject to} \sum_{i=1}^{I} \theta_i = 1 \ and \ \theta_s \in [0, 1],$$

where $f_n(\boldsymbol{\theta})$ is the $n$-th objective function, representing MAE (i.e., performance) of the candidate surveillance system defined by the design parameter $\boldsymbol{\theta}$.

The first objective function explored ($f_1(\boldsymbol{\theta})$) represents the MAE of the estimated serotype-specific incidence rates of all cases (i.e., incidence rates of EV-A71, CV-A16, and other enteroviruses) across locations, time, serotypes, and realizations (i.e., samples from the posterior distribution) of disease system model, expressed as:

$$f_1(\boldsymbol{\theta}) = \sum_{i=1}^{I} \sum_{t=1}^{T} \sum_{k=1}^{K} \sum_{r=1}^{R} \frac{|\lambda_{ikt}^{(r)} - \hat{\lambda}_{ikt|\boldsymbol{\theta}}^{(r)}|}{ITKR},$$

Where $I$, $T$, $K$, and $R$ represent the total number of locations ($I = 21$), study years ($T = 6$), serotypes ($K = 3$; for CA-V16 [$k = 1$], EV-A71 [$k = 2$], and other enterovirus [$k = 3$]), and disease system model realizations ($R = 80$, selected to ensure convergence of the estimated MAE

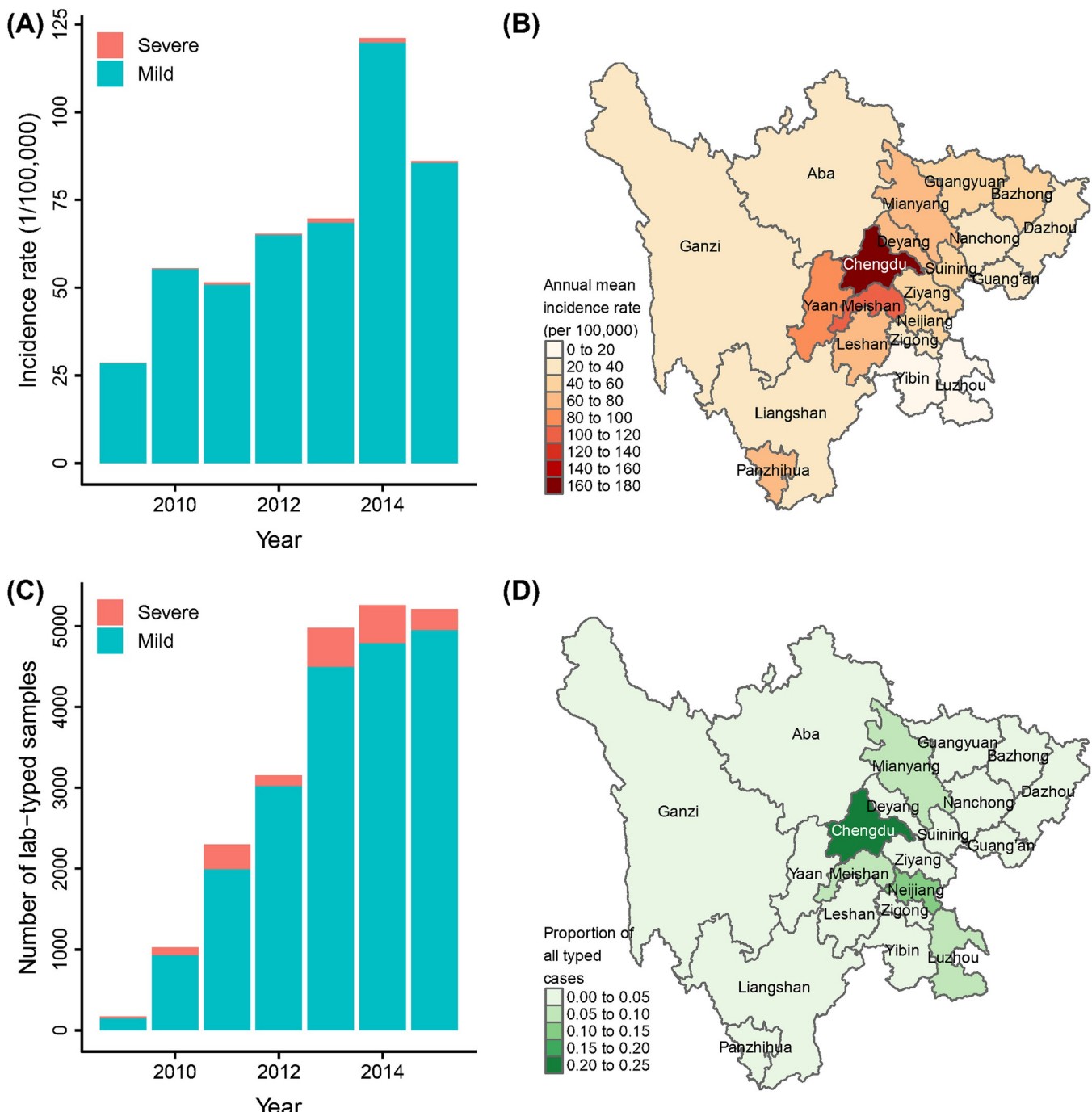

**Fig 2.** Temporal and spatial variations in HFMD incidence rate (A,B) and laboratory serotyping (C,D). (A) HFMD incidence rate for Sichuan 2009–2015; (B) annual mean HFMD incidence rate for each prefecture; (C) number of serotyped HFMD cases by year; (D) proportion of all serotyped cases drawn from each prefecture from 2009–2015. The boundaries of the prefectures were obtained from https://gadm.org/download_country.html.

across model runs), respectively; $\lambda_{ikt}^{(r)}$ represents the simulated incidence rate of the $i$th location during the $t$th year for the $k$th serotype in the $r$th realization of the disease system model; and $\hat{\lambda}_{ikt|\theta}^{(r)}$ represents the corresponding incidence rate estimated using the laboratory surveillance information ascertained by the surveillance system defined by the design parameter $\theta$. The

methods for simulating $\lambda_{ikt}^{(r)}$ and estimating $\hat{\lambda}_{ikt|\boldsymbol{\theta}}^{(r)}$ with HFMD surveillance data in the study province are described below in sections *2.2.4* and *2.2.5*, respectively.

An alternative objective function examined ($f_2(\boldsymbol{\theta})$) represented the MAE of the estimated serotype-specific incidence rates of severe cases across locations, time, serotypes, and realizations of disease system model, defined as:

$$f_2(\boldsymbol{\theta}) = \sum_{i=1}^{I} \sum_{t=1}^{T} \sum_{k=1}^{K} \sum_{r=1}^{R} \frac{|p_k^{(r)}\lambda_{ikt}^{(r)} - \hat{p}_{k|\boldsymbol{\theta}}^{(r)}\hat{\lambda}_{ikt|\boldsymbol{\theta}}^{(r)}|}{ITKR},$$

where $p_k^{(r)}$ represents the simulated probability of the $k$th serotype causing severe disease in the $r$th realization, while $\hat{p}_{k|\boldsymbol{\theta}}^{(r)}$ represents its estimate with information ascertained by the surveillance system defined by $\boldsymbol{\theta}$.

**2.2.4 Disease system model.** We estimated the underlying serotype-specific incidence rates in each region ($\lambda_{ikt}$) and the serotype-specific probability of severe disease ($p_k$) using data in the study region with a multivariate spatio-temporal Bayesian hierarchical framework (i.e., "the disease system model"; see schematic and hyperparameter priors in S2 Fig). The unobserved incidence rate of cases caused by serotype $k$, in location $i$, in year $t$, $\lambda_{ikt}$, is modeled as:

$$\lambda_{ikt} = \exp(\beta_0 + \boldsymbol{X}_{ikt}^{\tau}\boldsymbol{\beta}_{kt} + \gamma_{ikt}),$$

$$\gamma_{ikt} \sim \text{MultiVariateNormal}(0, \Sigma),$$

where $\beta_0$ represents the intercept; $\boldsymbol{X}_{ikt}$ represents disease risk factors with corresponding coefficients $\boldsymbol{\beta}_{kt}$ (although for simplicity, we incorporate only an intercept, but no risk factors, in the model); and $\gamma_{ikt}$ is a random effect. The vector of $\gamma_{ikt}$ is organized as $(\gamma_{111}, \cdots, \gamma_{I11}, \gamma_{121}, \cdots, \gamma_{IK1}, \gamma_{112}, \cdots, \gamma_{IKT})^{\tau}$ with a covariance matrix $\Sigma$, which is a separable multivariate space-time conditional autoregressive (MSTCAR) structure. More specifically, $\Sigma$ is the Kronecker product of three covariance matrices characterizing: the spatial dependence; between-serotype dependence; and the temporal dependence (see S2 Fig for details) [39].

Observed data, representing total HFMD cases at location $i$, in year $t$, with severity $s$, are denoted as $Y_{i.t}^{(s)}$, and serotyping results, $Z_{ikt}^{(s)}$, are used to infer the latent disease process parameters, as well as parameters of the observation process. Given the large population size of each location, the number of new cases in each location is assumed to be adequately represented by a Poisson distribution [40]:

$$Y_{i.t}^{(s)} \sim Poisson(\lambda_{i.t}^{(s)}N_{it}),$$

where $\lambda_{i.t}^{(s)} = \sum_{k=1}^{3}\lambda_{ikt}^{(s)}$ represents the aggregated incidence rate across serotype groups in location $i$, during year $t$, with severity $s$ ($s = 1$ represents severe disease; $s = 2$, mild disease); and $N_{it}$ represents the population size of location $i$ at year $t$. We denote the probability of serotype $k$ causing severe disease as $p_k$. Thus, the incidence rate of cases of severe disease is $\lambda_{ikt}^{(1)} = p_k\lambda_{ikt}$, while that of mild disease is $\lambda_{ikt}^{(2)} = (1 - p_k)\lambda_{ikt}$, and $\lambda_{ikt} = \sum_{s=1}^{2}\lambda_{ikt}^{(s)}$. We assume that the probability of being selected for laboratory typing does not depend on serotype after conditioning on case severity [2]. The number of annual tests is large, and thus test-positive case counts for each serotype are assumed to be adequately represented by a Poisson distribution:

$$Z_{ikt}^{(s)} \sim Poisson(\lambda_{ikt}^{(s)}N_{it}\phi_{it,test}^{(s)}),$$

where $Z_{ikt}^{(s)}$ represents the number of cases tested positive for serotype $k$ at location $i$, year $t$,

with severity $s$; and $\phi_{it,test}^{(s)}$ represents the probability of being selected for serotyping at location $i$, year $t$, with severity $s$, which was estimated by smoothing observed data in the study region by assuming spatial and temporal autocorrelation. Epidemiologic parameters estimated by the disease system model can be found in S1 Text.

*Generating disease data.* To ensure that our optimized surveillance scenarios account for uncertainty in the observation process and parameter estimates, after fitting the disease process and observation model to data from 2009–2014, R sets (R = 80) of serotype-specific incidence rates ($\lambda_{ikt}^{(r)}$) and parameter values (including $\beta_0$ and hyperparameters of $\Sigma$) were drawn from the joint posterior distributions. The sampled parameter values were used by the surveillance model (section *2.2.5*) to simulate $Z_{ikt|\theta}^{(s)}$ and to estimate $\hat{\lambda}_{ikt|\theta}^{(r)}$ and $\hat{p}_{k|\theta}^{(r)}$ under different surveillance designs.

**2.2.5 Surveillance model.** The surveillance model generates realizations of surveillance information conditional on the simulated disease data and the candidate design parameter. After proposing a sample allocation vector $\theta$, we first estimated the number of typing tests allocated to location $i$ in year $t$, for case severity s based on $\theta$ and the total number of laboratory typing tests conducted in year $t$ across all locations (Fig 2C), then further estimated $\phi_{it,test|\theta}^{(s)}$, the probability of being typed based on the estimated number of typing tests and the total observed number of HFMD cases at location $i$ and year $t$. The estimated probability $\phi_{it,test|\theta}^{(s)}$, together with the $r$th sample of $\beta_0$ and hyperparameters of $\Sigma$, were then used to re-estimate $\hat{\lambda}_{ikt|\theta}^{(r)}$ and $\hat{p}_{k|\theta}^{(r)}$ based on the disease system model described in section *2.2.4*, and to further evaluate the objective functions $f_1(\theta)$ and $f_2(\theta)$.

**2.2.6 Optimization search.** Since design vector $\theta$ is constrained by $\sum_{i=1}^{I} \theta_i = 1$, possibly rendering the optimization search process less efficient than an unconstrained optimization problem, we first converted the 22-dimensional design vector $\theta$ to an unconstrained 21-dimension internal design vector $\theta'$ by following methods described elsewhere [41]. This internal design vector $\theta'$ was then optimized with a genetic algorithm (GA)—a metaheuristic optimization algorithm inspired by a natural selection process [42]. GAs have the ability to handle complex optimization problems, avoid local optima, and find near-optimal solutions within a reasonable amount of time [43,44], and have been used extensively in public health and medical research [45–50].

To optimize with a GA, first an initial population of $n$ random designs was generated and the objective function value (i.e., MAE of estimated subtype-specific incidence rates, $f_1(\theta)$) of each design was evaluated. A small number of designs with the lowest MAEs survived to the next generation, while other designs were selected for recombination with probability determined by a function of their MAE. For each randomly matched pair of designs in the recombination pool, two new descendants were produced for the next generation, during which crossover occurs with high probability, $p_{crossover}$, and mutation occurs with low probability, $p_{mutation}$. If crossover occurs, the descendants were generated as linear combinations of the parent designs with randomly sampled weights. For example, if the two parents are $\theta_a'$ and $\theta_b'$, and the random weight sampled from Uniform(0,1) is $\omega$, then the two descendants are $\omega\theta_a' + (1 - \omega)\theta_b'$ and $(1 - \omega)\theta_a' + \omega\theta_b'$, respectively. When mutation happens, one random element of the design vector is changed to a random number sampled from its domain. Following previous studies [49,51], we set the initial population size n = 50, $p_{crossover}$ = 0.8, and $p_{mutation}$ = 0.05. The optimization process took about 45 hours on two nodes, each with a 96 GB RAM and two Skylake 20-core 2.1 GHz processors.

**2.2.7 Benchmarking and evaluation of robustness of optima.** The surveillance performance of the optimal design was benchmarked against seven archetypal designs: 1) the existing

allocation of laboratory typing across locations (Fig 2D, hereafter referred to as *Existing*); 2) an equal allocation of typing across all locations (hereafter *Equal*); 3) allocation of typing proportional to the location's population (hereafter *PopSize*); 4) allocation of typing proportional to absolute number of HFMD cases (hereafter *Case*); 5) allocation of typing proportional to HFMD incidence rate (hereafter *IncRate*); 6) allocation of typing proportional to absolute number of severe HFMD cases (hereafter *SevereCase*); 7) allocation of typing proportional to HFMD incidence rate of severe cases (hereafter *SevereIncRate*). See S3 and S4 Figs for the proportion of serotyping allocated to each location under each of these archetypal designs. The proportion of typing tests allocated to each location for these archetypal designs was estimated based on the 2009–2014 data, while the probabilities of severe cases serotyped were set to the values that minimize the MAEs with the corresponding locational allocation strategy, according to grid searches (S5 Fig). These seven archetypal designs were included in the initial population of the GA, together with another 43 randomly generated designs.

To examine the robustness of the designs selected by the optimization process, epidemiologic data for 2015 were held out to establish whether optimal designs based on 2009–2014 data performed well for the near-term future. During this process, we compared surveillance performance of the optimal design obtained with the 2009–2014 data to that of the seven archetypal designs described above, using only 2015 data. Furthermore, to investigate how robust the optimal design was when the total typing capacity changes, we repeated the analyses with halved, doubled, and quintupled total frequency of typing across all locations in each year (i.e., scaling the observed frequencies shown in Fig 2C). As an alternative method to examine if the performance of the surveillance system changes with resource limits, we also randomly selected 300 designs from the design space, evaluated the two objective values with the original resource constraint and when each constraint was halved, doubled, or quintupled, and investigated if the designs that performed well under one constraint also performed well under others.

**2.2.8 Computing platform and code availability.** All analyses were conducted in R 4.0.3 [52] on Berkeley's Savio computational cluster [53], with *rstan* package 2.18.2 for Bayesian hierarchical modeling [54], *GA* package 3.2 for implementation of the genetic algorithm [55], and packages *ggplot2* 3.1.1 [56], *cowplot* 0.9.4 [57], and *tmap* 2.2 [58] for visualization. All code and data are available at: https://github.com/qu-cheng/Lab_surveillance_optimization

## 3 Results

### 3.1 Optimal designs

**Allocation by location.** The existing laboratory surveillance network (*Existing* archetypal design) allocates approximately a quarter of all subtyping effort to the most populous prefecture in the study region (Chengdu), while in contrast, less than one percent of subtyping effort is allocated to Ganzi, a remote prefecture in the northwestern mountainous region of the study area (Figs 3A and 2D). The optimal designs to minimize error in estimated serotype-specific incidence rates of all HFMD cases (*Optimal* for all) and only severe HFMD cases (*Optimal* for severe) shift the typing allocation substantially (Figs 3C and 3D and S6). Although the very populous Chengdu prefecture still receives the largest proportion of typing resources, the optimal designs allocate just 12.2% and 9.5% of total typing resources for the two objectives, respectively. Notably, in S7 Fig, which shows the proportion of cases being serotyped at each location according to the *Optimal for all* and *Optimal for severe* designs, certain prefectures with low absolute typing allocations (e.g., Ganzi and Aba in Fig 3C and 3D) are able to serotype a large proportion of total cases (e.g., >30 percent of cases are serotyped in Ganzi and Aba); for the populous Chengdu prefecture, optimal designs serotyped less than 2% of total cases in this prefecture, by comparison.

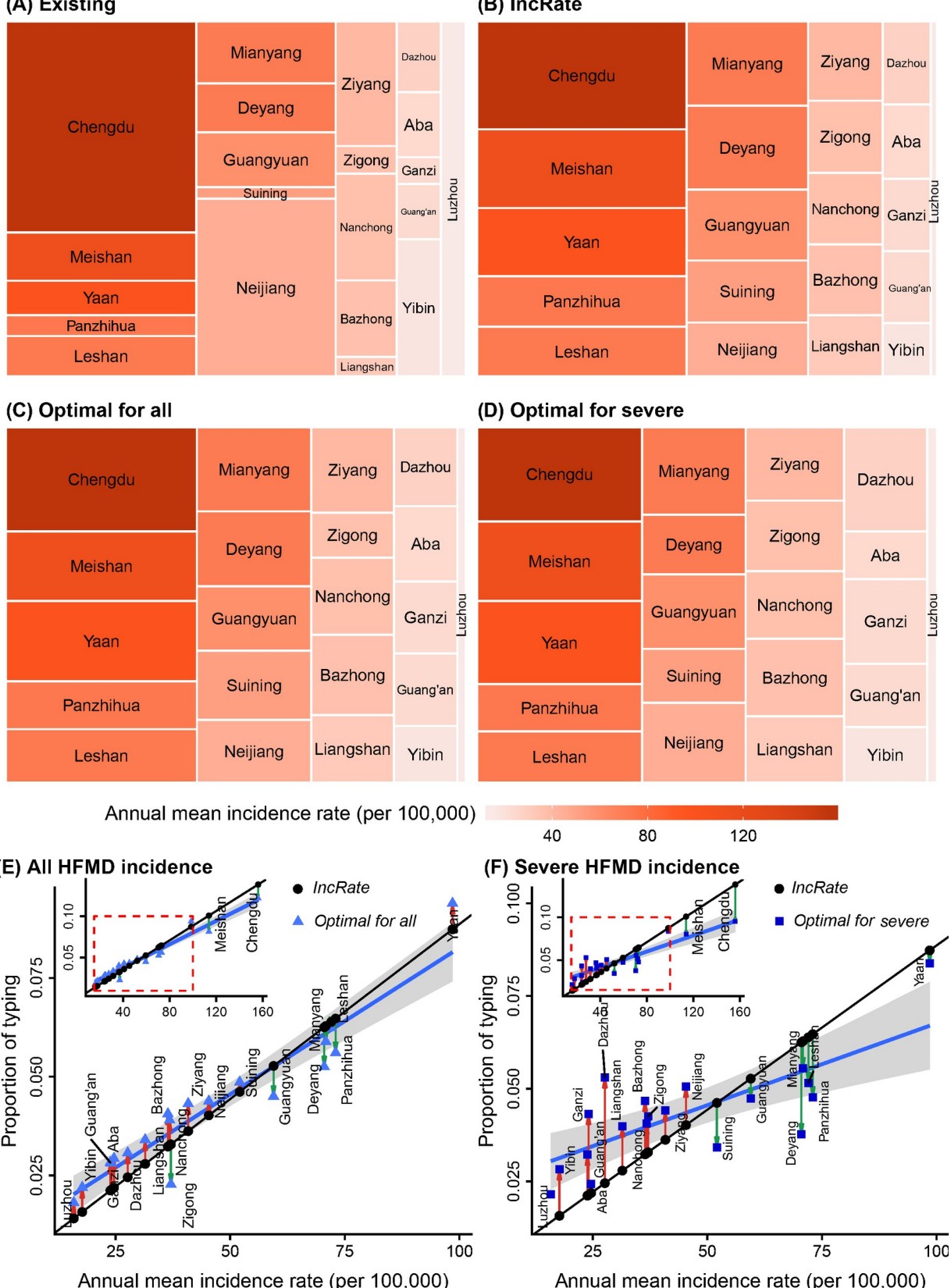

**Fig 3. Comparison between *Existing*, *IncRate*, and *Optimal* subtyping allocation strategies across locations.** Treemaps show the proportion of typing efforts allocated to each location in the (A) *Existing*, (B) *IncRate*, and *Optimal* designs that minimize the error in estimated serotype-specific incidence rate of (C) all HFMD cases and (D) only severe HFMD cases. Tiles represent study locations, with the area of the tile representing the proportion of all typing efforts allocated to the location, and the color of the tile representing the location's annual mean HFMD incidence rate. Tiles are ordered by decreasing annual mean incidence rate from top to bottom, then left to right. Scatterplots show the correlation between annual mean incidence rate of the optimal proportion of total typing resources allocated to each location to minimize error in estimated serotype-specific incidence rate of (E) all HMFD cases and (F) only severe HFMD cases. Black dots represent the archetypal design *IncRate* (see definition in section 2.2.7), blue triangles in (E) and squares in (F) represent the optimal allocation strategy for minimizing error in estimating serotype-specific incidence rate for all cases and only severe cases, respectively. The blue lines represent the best fit relating annual mean incidence rates to typing allocations across the *Optimal* designs. Vertical arrows represent changes from *IncRate* to *Optimal*: red arrows represent increases in typing efforts from *IncRate* to *Optimal*; green arrows represent reductions in typing efforts from *IncRate* to *Optimal*. Inset figures show data for all prefectures, showing the range (red dashed rectangle) displayed in the main panel.

The designs *Optimal for all* and *Optimal for severe* are similar to the archetypal design *IncRate* (compare Fig 3B with Fig 3C and 3D). Moreover, the optimal proportions of total typing resources to allocate to each location for both surveillance objectives are correlated with the annual mean incidence rate of the location (Fig 3E and 3F), although the typing efforts are more equally distributed in the *Optimal for severe* design than in the *Optimal for all* design (compare locations across Fig 3C and 3D, and the difference in slope of the blue lines in Fig 3E and 3F).

**Allocation by case severity.** The optimized proportion of severe cases to serotype depended strongly on the surveillance objective: 0.17 when minimizing errors in serotype-specific total HFMD incidence rates, and 0.70 when minimizing errors in serotype-specific severe HFMD incidence rates. To explore the effect of changing the proportion of severe cases being serotyped on surveillance performance, we fixed the spatial allocation of typing resources for each objective at the values in the *Optimal* designs while varying the proportion of severe cases subjected to serotyping from 0.01 to 0.99. The mean absolute error (MAE) of estimated total serotype-specific HFMD incidence rates was minimized at 11% of severe cases serotyped (Fig 4A). Notably, the MAE increases for this goal as severe cases are increasingly prioritized for serotyping.

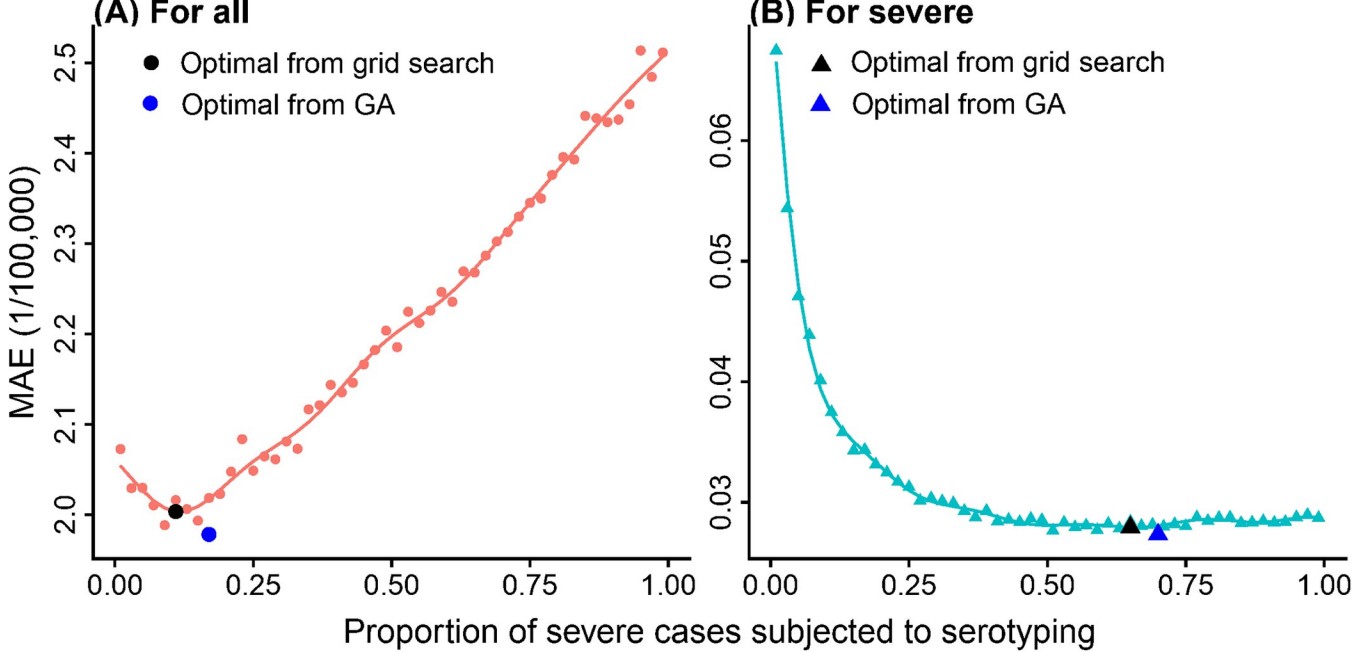

**Fig 4.** Impact of the proportion of severe cases serotyped on mean absolute error (MAE) of the estimated serotype-specific incidence rate of (A) all HFMD cases and (B) severe HFMD cases. Colored lines are smoothed by Gaussian process models. Black dot and triangle represent the probabilities of severe cases being serotyped that lead to the lowest error in estimating serotype-specific incidence rate of all (dot) and only severe (triangle) HFMD cases; blue dot and triangle represent the optimal designs from GA.

For severe HFMD cases, the MAE initially decreases as greater proportions of severe cases are serotyped, then plateaus when about half of the severe cases are serotyped, reaching its optimum when the proportion of severe cases serotyped is 0.65. The optimal proportion of severe cases subjected to serotyping identified by GA are very close to the ones identified in this experiment, which suggests that the GA successfully explored the design space. For further analyses, we updated the probability of serotyping severe cases in both *Optimal* designs to be the values identified in this grid search of $\theta_s$ conditioning on optimal values of $\theta_1, \theta_2, \ldots, \theta_I$ found by the GA, as the conditional grid search guarantees a better or equal estimate of $\theta_s$.

### 3.2 Comparisons with archetypal designs

The optimal allocation of subtyping among regional subpopulations and case severity groups —while adhering to the same level of typing effort as the current design (*Existing*)—yielded a significant improvement in estimating the target parameters. The distribution of error (MAE) of estimated serotype-specific incidence rate of all HFMD and severe HFMD cases, across location, serotype, and year in 1000 realizations of the disease model for the optimal design was compared to the seven archetypal designs described in section *2.2.7* (Fig 5). When compared with the current surveillance design (*Existing*), with the same number of cases subjected to serotyping, the selected optimal designs (*Optimal*) exhibit 14.1 and 20.5 percent lower average MAE for the estimated serotype-specific incidence rate of all cases for the 2009–2014 (Fig 5A) and 2015 (Fig 5C) period, respectively; and a 13.3 and 14.8 percent lower average MAE of the estimated serotype-specific incidence rate of only severe cases for the 2009–2014 (Fig 5B) and 2015 (Fig 5D) period, respectively. Among the archetypal designs, *IncRate* generally performed well for both objectives. The results indicate that optimal designs based on historical observed data from 2009–2014 performed well for the 2015 year, which was held out of the optimization procedure, suggesting that optimal designs identified by DIOS may be useful for planning typing resource allocations in the short-term future.

### 3.3 Sensitivity of selected designs to the total number of cases sampled for subtyping

**Allocation by location.** To investigate whether the optimal design is robust to changes in the availability of typing resources, we compared the optimal designs for both objectives when the frequency of typing is set to half, two times, or five times that of historical serotyping rates. With more typing resources, MAE of estimated serotype-specific incidence rate of total and severe cases decreases substantially (S8 Fig), while the optimal location-wise allocation changes modestly (Figs 6 and S9 and S10). For both surveillance objectives, as serotyping resources increase, the optimal proportion of typing allocated at each location tends to become more evenly distributed, particularly for estimating the serotype-specific incidence rates of severe HFMD, because the marginal benefits of more intensive serotyping at locations with higher incidence fall, while more frequent serotyping at locations with lower incidence rates can continue to reduce estimation error.

When examining 300 randomly sampled designs, the MAE of estimated serotype-specific incidence rate of total and severe cases across the four resource limit scenarios were highly correlated ($>0.8$, S11 Fig), which again suggests that the optimal allocation of laboratory resources is relatively insensitive to resource constraints in this framework, even as additional typing resources results in lower estimation errors.

**Allocation by case severity.** When seeking to minimize error in estimated serotype-specific incidence rates of all HFMD cases, the optimal proportion of severe cases to serotype decreases as the availability of typing resources increases (Fig 7A). Conversely, when seeking

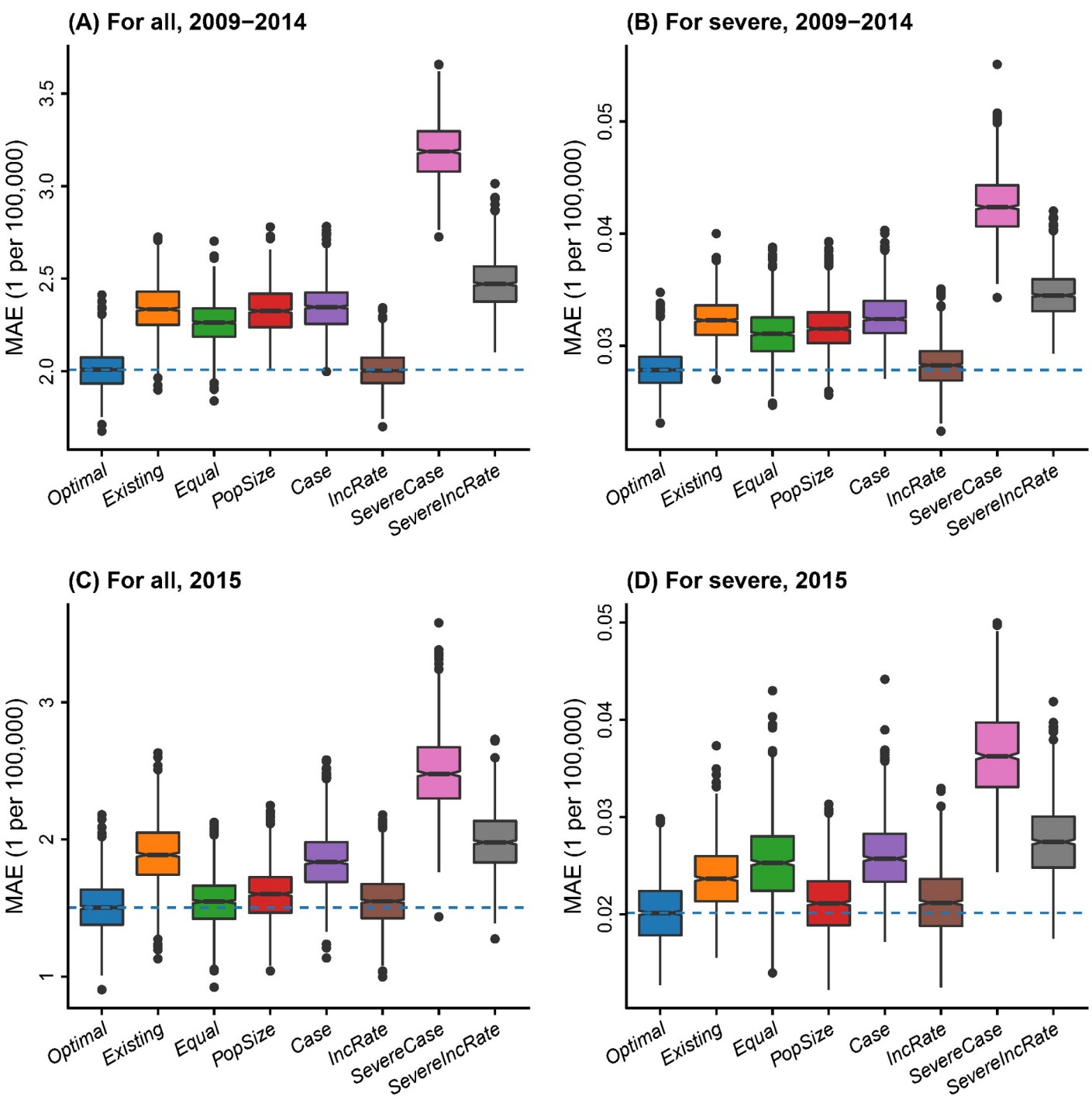

**Fig 5. Surveillance performance of the optimal design and the seven archetypal designs evaluated with data from 2009–2014 and 2015 over 1000 realizations of the disease system model.** Violin plots and boxplots for different designs (shades of color) show the distribution of mean absolute error (MAE) in estimating serotype-specific incidence rates of (A) all cases and (B) only severe cases using 2009–2014 data; and (C) all cases and (D) only severe cases using 2015 data, which was not used in the optimization procedure. The horizontal dashed lines show the median MAEs of the optimal designs.

to minimize error in estimated serotype-specific incidence rates of *severe* HFMD cases, the optimal proportion of severe cases to serotype increases as the availability of typing resources increases (Fig 7B). This is likely because for estimating the serotype-specific incidence rates of

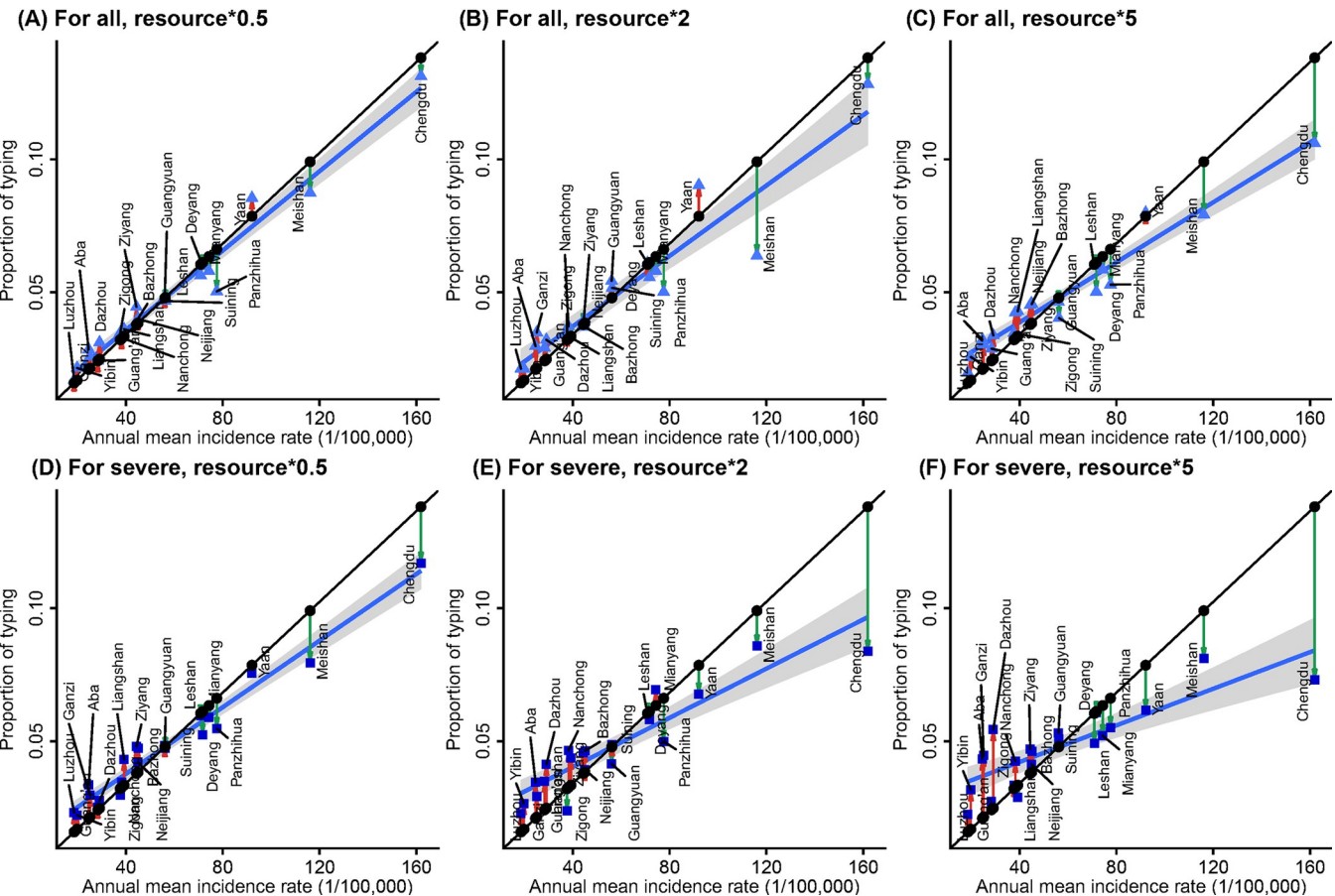

**Fig 6.** Scatterplots of annual mean incidence rate and the proportion of typing resources allocating to each location under the archetypal design IncRate (black dots) and the Optimal designs for minimizing the MAE of estimated serotype-incidence rate of all HFMD cases (blue triangles) when the available typing resources is (A) halved, (B) doubled, and (C) quintupled; and the Optimal designs for minimizing the MAE of estimated serotype-incidence rate of severe HFMD cases (blue squares) when the available typing resources is (D) halved, (E) doubled, and (F) quintupled.

all HFMD cases, the marginal improvements diminish as long as enough samples of severe cases are tested to accurately estimate the virulence of each serotype; while for estimating the serotype-specific incidence rates of severe HFMD cases, the errors continue to decrease as more severe cases are tested.

## 4 Discussion

Laboratory-based disease surveillance networks are often designed in an *ad hoc* manner, guided by budgetary, logistical, or infrastructural considerations [19], which may lead to inefficient use of limited typing resources. Here, we adapted the DIOS framework to provide a quantitative platform for the simulation of epidemiologic and surveillance processes in the context of optimizing the allocation of scarce laboratory typing resources under operational constraints. In a case study, we apply the framework to determine how a limited number of samples for typing should be drawn from subpopulations to optimize estimation of serotype-specific incidence rates for all—and the subset of severe—HFMD cases in a study region in China.

We demonstrated that, with the same level of typing effort as the existing network, optimal designs chosen using DIOS can reduce the mean absolute error of estimates of serotype-

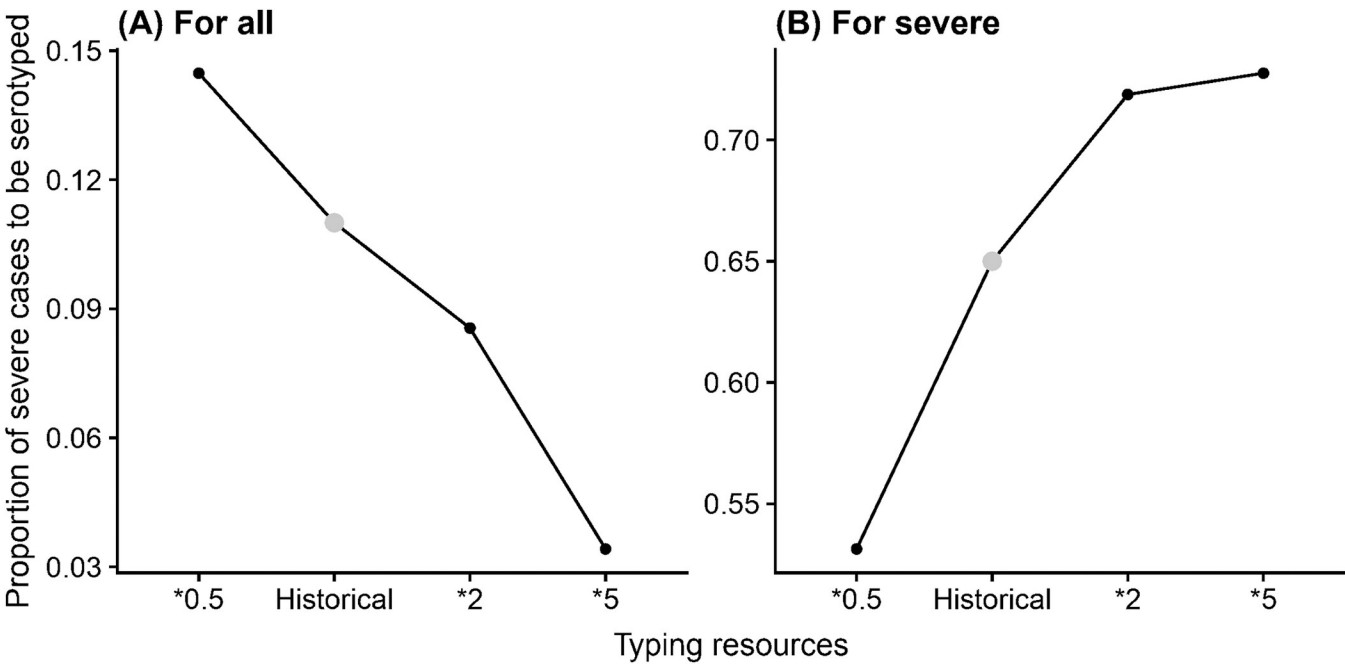

**Fig 7.** Optimal proportion of severe cases to be subjected to serotyping as the availability of typing resources changes, when seeking to minimize error serotype-specific incidence rates of (A) all HFMD cases and (B) only severe HFMD cases.

specific incidence rates and proportions of clinical cases caused by each serotype by 14.1 and 13.3 percent, respectively. Although beyond the scope of this study, the DIOS framework accommodates multi-objective optimization as well [20], providing a means to identify optimal designs for simultaneously optimizing both objectives. Changes to the total number of cases sampled for subtyping minimally impacted the relative performance of surveillance designs.

Our optimization identified that allocating laboratory typing resources across locations in proportion to their HFMD incidence rates gave near-optimal performance for estimating both the total serotype-specific incidence rates and serotype-specific incidence rates of severe HFMD. For estimating total HFMD incidence, this is fairly intuitive, since errors in incidence rates will exhibit higher variance when the incidence rate itself is higher, additional typing to stabilize these estimates across locations will benefit the average MAE. The optimal design for estimating serotype-specific incidence rates of severe cases involves a slightly more equal distribution of subtyping resources, in part because fewer tests are available to type mild cases as the proportion of severe cases typed increases in the optimal design, which results in insufficient resources to accurately estimate background serotype-specific incidence of mild cases at locations with low incidence rates.

Our study opens several areas for future research. While we focused on a surveillance design parameter representing the proportion of all typed cases to be drawn from each region, other design parameters can certainly be examined, such as the sampling of cases for subtyping across demographic groups, the selection of laboratories to include in the surveillance network, and the assays used for typing. Besides the total number of typing tests, other constraints, such as the total cost for processing and shipping the specimens given the fact that the cost may vary across locations, can also be considered. What is more, other surveillance objectives—beyond estimating serotype-specific incidence rates for all cases and only severe cases—are possible, such as early detection of a new subtype or an unusual increase in existing

subtypes, evaluation of the effectiveness of subtype-specific interventions, and confirmation of the elimination or eradication of a specific subtype. Multiple objectives can also be evaluated simultaneously through multi-objective optimization, as we have demonstrated elsewhere [20].

Expanding on the use of a single disease system model here, multiple models with different structures—e.g., hierarchical models with different covariance structures, machine learning algorithms, and mechanistical models—and different parameter values or covariates could be run in an ensemble to better represent uncertainty in the underlying epidemiologic processes. Periodic intensive, cross-sectional sampling may also help to validate and fine-tune the design optimization process by providing high-resolution, high-confidence estimates of incidence rates. Furthermore, while this study assumed that the optimal design is fixed and does not change over time, future optimizations could update optimal designs iteratively as new data becomes available, refitting the disease system model and updating the optimal design. Such an adaptive sampling approach may result in improved surveillance performance in settings where transmission dynamics change substantially over time [59].

In conclusion, we have shown that designing laboratory networks for surveillance systems with the DIOS framework can reveal designs that allocate limited resources more efficiently. For jurisdictions with sophisticated computational capabilities, the analyses in this work could be repeated to identify the optimal designs for specific settings and surveillance goals. For regions with limited resources, rules of thumb, such as the allocation of typing resources in proportion to incidence rates, may emerge from simulations of general scenarios. Future work is needed to generate such transcendent surveillance rules for various surveillance design parameters and goals, and to yield improved understanding of the design parameters that would allow the most cost-effective laboratory-based surveillance architectures. The scope of applications of the DIOS framework extends across many dimensions of laboratory-based surveillance networks and associated goals, raising important opportunities for developing the next generation of laboratory surveillance systems to monitor pathogen subtypes.

## Supporting information

**S1 Text. Epidemiological parameter estimates by the disease system model.**
(DOCX)

**S1 Fig.** Annual mean percentage of cases being tested for (A) all clinical HFMD cases, (B) mild cases, and (C) severe cases between 2009–2015. The boundaries of the prefectures were obtained from https://gadm.org/download_country.html.
(PDF)

**S2 Fig. Schematic of the multivariate spatio-temporal Bayesian hierarchical model.** See the main text for the definitions of notations. Priors of the hyperparameters are highlighted in blue, while observed data are highlighted in green.
(PDF)

**S3 Fig.** Proportion of typing resources allocate to each location for the archetypal designs: (A) *Existing*, (B) *Equal*, (C) *PopSize*, (D) *Case*, (E) *IncRate*, (F) *SevereCase*, and (G) *SevereIncRate*. See descriptions of these designs in Section 2.2.7 of the main text. The prefectures are colored by the proportion of serotyping resources allocated to them, with darker colors representing more serotyping resources. The boundaries of the prefectures were obtained from https://gadm.org/download_country.html.
(PDF)

**S4 Fig.** Proportion of serotyping resources allocate to each location for the archetypal designs: (A) *Existing*, (B) *Equal*, (C) *PopSize*, (D) *Case*, (E) *IncRate*, (F) *SevereCase*, and (G) *SevereIncRate*. See descriptions of these designs in Section 2.2.7 of the main text. Each tile represent one location, with the area of the tile proportional to the amount of typing resources allocated to it and the color of the tile representing the annual mean incidence rate of that location.
(PDF)

**S5 Fig.** Optimal probability of severe cases being serotyped for each archetypal design minimizing mean absolute errors (MAE) of the estimated serotype-specific incidence rate of (A) all HFMD cases and (B) only severe HFMD cases. Different colors represent different archetypal designs. The colored lines are smoothed by Gaussian Process model. Black dots and triangles represent the optimal probability of severe cases being serotyped for each archetypal design minimizing mean absolute errors (MAE) of the estimated serotype-specific incidence rate of all HFMD cases and only severe HFMD cases, respectively.
(PDF)

**S6 Fig.** The optimal proportion of subtyping to allocate to each location for minimizing mean absolute error in estimating serotype-specific incidence rate of (A) all cases and (B) severe cases. The boundaries of the prefectures were obtained from https://gadm.org/download_country.html.
(PDF)

**S7 Fig.** The proportion of cases being subtyped according to the optimal designs that minimize mean absolute error in estimating serotype-specific incidence rate of (A) all cases and (B) severe cases. The boundaries of the prefectures were obtained from https://gadm.org/download_country.html.
(PDF)

**S8 Fig.** Mean absolute error in estimating serotype-specific incidence rate of (A) all cases and (B) only severe cases when the availability of typing resources changes.
(PDF)

**S9 Fig.** The optimal proportion of subtyping to allocate to each location for minimizing mean absolute error in estimating serotype-specific incidence rate of all cases when the total amount of subtyping resources is (A) half, (B) two times, or (C) five times that of the observed frequency; and for minimizing mean absolute error in estimating serotype-specific incidence rate of severe cases when the total amount of subtyping resources is (D) half, (E) two times, or (F) five times that of the observed frequency. The boundaries of the prefectures were obtained from https://gadm.org/download_country.html.
(PDF)

**S10 Fig.** Scatterplots of annual mean incidence rate and the proportion of typing resources allocating to each location under the archetypal design IncRate (black dots) and the Optimal designs for minimizing the MAE of estimated serotype-incidence rate of all HFMD cases (blue triangles) when the available typing resources is (A) halved, (B) doubled, and (C) quintupled; and the Optimal designs for minimizing the MAE of estimated serotype-incidence rate of severe HFMD cases (blue squares) when the available typing resources is (D) halved, (E) doubled, and (F) quintupled.
(PDF)

**S11 Fig. Correlation between the objective function values under four resource limit scenarios.** Correlation between the MAEs of estimated serotype-specific incidence rate of (A) all

cases and (B) only severe cases.
(PDF)

## Acknowledgments

This research benefitted from the Savio computational cluster resource provided by the Berkeley Research Computing program at the University of California, Berkeley, which is supported by the UC Berkeley Chancellor, Vice Chancellor for Research, and Chief Information Officer.

## Author Contributions

**Conceptualization:** Qu Cheng, Philip A. Collender, Alexandra K. Heaney, Aidan McLoughlin, Jennifer R. Head, Changhong Yang, Justin V. Remais.

**Data curation:** Qu Cheng, Rohini Dasan, Song Liang, Qiang Lv, Yaqiong Liu, Changhong Yang.

**Formal analysis:** Qu Cheng, Philip A. Collender.

**Funding acquisition:** Justin V. Remais.

**Investigation:** Qu Cheng, Philip A. Collender.

**Methodology:** Qu Cheng, Philip A. Collender, Yang Yang, Yuzi Zhang, Howard H. Chang, Lance A. Waller, Jon Zelner, Joseph A. Lewnard.

**Supervision:** Justin V. Remais.

**Visualization:** Qu Cheng.

**Writing – original draft:** Qu Cheng.

**Writing – review & editing:** Qu Cheng, Philip A. Collender, Alexandra K. Heaney, Aidan McLoughlin, Yang Yang, Yuzi Zhang, Jennifer R. Head, Rohini Dasan, Song Liang, Qiang Lv, Yaqiong Liu, Changhong Yang, Howard H. Chang, Lance A. Waller, Jon Zelner, Joseph A. Lewnard, Justin V. Remais.

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
