## [Decision Letter · Decision Letter 0]

23 May 2022

Dear Prof. Remais,

Thank you very much for submitting your manuscript "Optimizing laboratory-based surveillance networks for monitoring multi-genotype or multi-serotype infections" for consideration at PLOS Computational Biology. As with all papers reviewed by the journal, your manuscript was reviewed by members of the editorial board and by several independent reviewers. The reviewers appreciated the attention to an important topic. Based on the reviews, we are likely to accept this manuscript for publication, providing that you modify the manuscript according to the review recommendations.

Sincerely,

Benjamin Althouse

Associate Editor

PLOS Computational Biology

Nina Fefferman

Deputy Editor

PLOS Computational Biology

Jason A. Papin

Editor-in-Chief

PLOS Computational Biology

Feilim Mac Gabhann

Editor-in-Chief

PLOS Computational Biology

[LINK]

Reviewer's Responses to Questions

**Comments to the Authors:**

Reviewer #1: General:

This is a Research Article submission by Cheng and colleagues focused on development of an analytic framework to improve decisions on sampling for infectious disease surveillance with the case example of Hand-foot-and-mouth disease in China. They model where to allocate a fixed number of PCR assays that identify viral etiology (CA-V16, EV-A71, or other) amongst N=21 locations with a goal of optimizing serotype specific case incidence. Overall this is an interesting analysis. My main concern is whether the data can truly support the authors’ goals.

Major comments:

-The goal of the analysis is to compare a model-driven sampling scheme to current practice (archetypal). Therefore a reference standard is needed to identify the “true” serotype specific incidence, by which to compare these approaches. Ideally, this would be high resolution sampling of all locations. However, the data from almost all locations except Chengdu appears quite limited (Figure 2D). Therefore, even with the approach done by the authors with resampling, I worry the data is limited to make an informed comparison. Would welcome clarification and input from the authors, and additional description of the available data.

Minor comments:

-I am not clear how the DIOS framework is particularly unique beyond an iterative algorithm for this case example.

-Could the authors clarify the “Realizations” from equation on line 269 and why there are 80?

-There appears to be the assumption that the number of samples is fixed rather than variable, which could be further explored.

-How do the authors deal with uncertainty?

-The authors mention additional dimensions of time and serotype, is varying these for optimization explored? Would be reasonable to do only location, just curious.

-Methods section is somewhat hard to follow.

Reviewer #2: Thank you for the chance to review the research article submission "Optimizing laboratory-based surveillance networks for monitoring multi-genotype or multi-serotype infections". This paper is a clear, well written, original, innovative and potentially rather important contribution to the literature on disease surveillance system design, evaluation and optimization. While the paper may appear to overlap in-part with the prior publication by Cheng et al in PLOS Comp Biol from 2020, https://journals.plos.org/ploscompbiol/article?id=10.1371/journal.pcbi.1008477, the current submission presents a very nice methodological advance in this area.

A few issues with editing and clarity are outlined below (by page and line number),

Page 5, Line 98-99: re. "and only 2-3 influenza cases are required to be typed..." Is this a typo, and should be indicating 2-3% of influenza cases, or does CDC and APHL actually say "2-3 influenza cases", and if so, I gather they also stipulate some denominator, possibly such as 2-3 cases per specified geography or time period or population.

P6 L110-118: re. "biases arising from sampling clinical cases for subtyping..." This language seems to imply that sampling follows a scheme that is intentionally designed in most cases, but I worry that it is the absence of design that is actually more typical. The academic and engineering perspective should clearly be informing hospital, clinical, laboratory and public health systems and practices. I just wonder if the assumptions in this paper are too far from reality, and a more basic and simple set of guidelines need to be met first by public health surveillance authorities.

P7 L138-143: re. "Applying DIOS involves specifying surveillance objectives..." Again, this may be assuming that surveillance systems are being developed, implemented, and are operating under more optimistic conditions then they actually are. While it is not the place for this paper to necessarily address or try to resolve the deficiencies of public health systems, it might be useful to recognize how far from optimal such practices are. To meaningfully run DIOS in most public health settings would likely require a major investment and a realignment in approach.

P10 L172-174: very true re. dynamics, and that it should represent known biases due to subtype interaction, but also possibly unrecognized interactions between variants/subtypes that that are not (yet) known.

P26 L457 and L460: possibly redundant, should it just be "GA" instead of "GA algorithm" here?

**Have the authors made all data and (if applicable) computational code underlying the findings in their manuscript fully available?**

Reviewer #1: Yes

Reviewer #2: Yes

PLOS authors have the option to publish the peer review history of their article (what does this mean?). If published, this will include your full peer review and any attached files.

Reviewer #1: No

Reviewer #2: No

Figure Files:

Data Requirements:

Reproducibility:

References:

---

## [Decision Letter · Decision Letter 1]

15 Sep 2022

Dear Prof. Remais,

We are pleased to inform you that your manuscript 'Optimizing laboratory-based surveillance networks for monitoring multi-genotype or multi-serotype infections' has been provisionally accepted for publication in PLOS Computational Biology.

Best regards,

Benjamin Althouse

Academic Editor

PLOS Computational Biology

Nina Fefferman

Section Editor

PLOS Computational Biology

Jason A. Papin

Editor-in-Chief

PLOS Computational Biology

Feilim Mac Gabhann

Editor-in-Chief

PLOS Computational Biology

Reviewer's Responses to Questions

**Comments to the Authors:**

Reviewer #1: Thank you for your revision.

**Have the authors made all data and (if applicable) computational code underlying the findings in their manuscript fully available?**

Reviewer #1: None

PLOS authors have the option to publish the peer review history of their article (what does this mean?). If published, this will include your full peer review and any attached files.

Reviewer #1: No

---

## [Editor Report · Acceptance letter]

22 Sep 2022

PCOMPBIOL-D-21-01940R1 

Optimizing laboratory-based surveillance networks for monitoring multi-genotype or multi-serotype infections

Dear Dr Remais,

I am pleased to inform you that your manuscript has been formally accepted for publication in PLOS Computational Biology. Your manuscript is now with our production department and you will be notified of the publication date in due course.

With kind regards,

Olena Szabo
